

# Transport, mixing, and feedback of dust, biomass burning and anthropogenic pollutants in eastern Asia: A case study

Derong Zhou[1], Ke Ding[1], Xin Huang[1,2,*], Lixia Liu[1,#], Qiang Liu[1], Zhengning Xu[1], Fei Jiang[3], Congbin Fu[1,2], Aijun Ding[1,2,*]

[1]Joint International Research Laboratory of Atmospheric and Earth System Sciences and School of Atmospheric Sciences, Nanjing University, Nanjing 210023, China

[2]Collaborative Innovation Center of Climate Change, Jiangsu Province, China

[3]Jiangsu Provincial Key Laboratory of Geographic Information Science and Technology, International Institute for Earth System Science, Nanjing University, Nanjing 210023, China.

[#]Now at Max Planck Institute for Chemistry, Mainz, Germany

*Correspondence to :* X. Huang (xinhuang@nju.edu.cn) or A. Ding (dingaj@nju.edu.cn)

**Abstract**

Anthropogenic fossil fuel (FF) combustion, biomass burning (BB) and desert dust are main sources of air pollutants around the globe. The emission of the three sources in Asia are all very intensive and their influences on air quality is very important, especially in spring. In this study, we investigate the vertical distribution, transport characteristics, source contribution, and meteorological feedback of the dust, BB and FF aerosols in a unique pollution episode occurred in eastern Asia based on various measurement data and modelling methods. Ground-based observations indicated a persistent pollution episode dramatically changing from secondary fine particulate pollution to dust pollution in late March 2015 over the Yangtze River Delta (YRD) region, eastern China. The online-coupled meteorology-chemistry-aerosol modelling together with Lagrangian particle dispersion simulations were conducted to investigate the vertical structure, transport characteristics and mechanisms of the multi-scale, multi-source, and multi-day air pollution episode. The regional polluted continental aerosols mainly accumulated near surface by local anthropogenic emissions mixed with dust aerosols, downwash from the upper planetary boundary layer (PBL) and middle/lower troposphere (MLT), and further transported downwardly by large-scale cold fronts and warm conveyor belts. BB smoke from the Southeast Asia, mainly from forest burning in Indochina, were transported by westerlies around the altitude of 3 km from southern China to eastern China, further mixed with dust and FF aerosols in eastern China and experienced long-range transport over the subtropical Pacific Ocean. The three pollutant sources could all transport





to eastern China, especially the YRD region around the latitude of 30°N, caused a structure of multi-layer pollutants and well mixed pollutants there. These solar absorption aerosols from FF, BB and dust could also cause significant feedback with MLT meteorology and then enhance local anthropogenic pollution. This study highlights the importance of intensive vertical measurement in the eastern China and the downwind Pacific Ocean with a focus of understanding the complex physical and

5   chemical processes of various pollution sources, and also raises the needs of quantitative understanding of environmental and climate impacts of these pollution sources in regional even global scales.

**Key words:** Anthropogenic aerosols, biomass burning, dust, vertical distribution, long-range transport, synoptic weather, meteorological feedback



# 1 Introduction

With rapid economic development and tremendous energy consumption in the past decades, East Asia, especially eastern China, undergoes increasingly severe air pollution (Zhang et al., 2012; van Donkelaar et al., 2010; Ding et al., 2016). The air pollutants in this region are very complicated because of not only dense and intense anthropogenic activities, but also multiple

natural pollution sources (like windblown dust, biomass burning (BB) and biogenic emissions) and a complex monsoon climate. The unique natural monsoon climate and strong human perturbations make the East Asia one of the unique regions to study the interactions between atmospheric physical and chemical processes, especially in the mixed anthropogenic and natural pollutants (Ding et al., 2017).

Dust is one of the most important aerosols influencing air quality and regional climate in eastern Asia even other

continents (Huang et al., 2014; Nie et al., 2014). In cold seasons, especially in spring, dust could transport from the inner Asian continent, such as Taklimakan and Gobi Deserts, and pass over the eastern China to the Pacific even the North America (Zhang et al., 2010; Liu et al., 2016) by the strong Asian monsoon together with the westerlies. These dust aerosols are usually mixed with anthropogenic pollutants along its transport pathways (Mori et al., 2003; Huang et al., 2010; Nie et al., 2014; Huang et al., 2014), resulting in complex interactions between physical and chemical processes and even meteorological feedbacks (Nie

et al., 2014; Xie et al., 2015; Liu et al., 2016; Yang et al., 2017). Based on field measurement at a mountain site in South China and the Station for Observing Regional Processes of the Earth System (SORPES) in eastern China, respectively, Nie et al. (2014) and Xie et al. (2015) reported that the mixed dust and anthropogenic pollutants promoted new particle formation and growth via heterogeneous photochemical chemistry. These newly formed aerosols, particularly sulfate particle, could further influence cloud condensation nuclei (CCN) over the downwind regions (Nie et al., 2014). Dust aerosols and anthropogenic

black carbon aerosols (soot) could influence the dynamics of planetary boundary layer (PBL) through radiative perturbations and subsequent impact on energy balance of the Earth-atmosphere system (Ding et al., 2016; Liu et al., 2016; Petäjä et al., 2016; Yang et al., 2017). The mixing of soot or dust with scattering aerosol components have strong direct and indirect impacts on radiation transfer (Li et al., 2011; Wu et al., 2016). The thermal effect changed dynamics in the lower troposphere in turn could influence the air pollutant dispersion and accumulation in the megacities, which further influence the vertical distribution

of air pollutants (Ding et al., 2016; Ding et al., 2017; Yang et al., 2017). Therefore, the study of transport, mixing and feedback processes/mechanisms of dust and anthropogenic pollutants is very important for improving current understandings of air pollution and its interactions with regional climate in eastern Asia. In early spring, strong continental outflow of biomass burning smoke particles and gases from Indochina also play an important role in increasing the peak values of trace gases in the MLT of eastern China (Hsu et al., 2003; Jacob et al., 2003; Zhou et al., 2013; Dong et al., 2015; Cohen et al., 2017). BB

plumes from Indochina could transport up the terrain through the westerlies to the southeast China and even to the Pacific Ocean (Jacob et al., 2003; Reid et al., 2013; Lin et al., 2014). The long-range transported plumes not only affected air quality at the ground surface in the downwind region, the uplifted plumes in the MLT could also change the structure of atmospheric components and meteorological parameters over the downwind areas in lower latitudes and the Northwest Pacific in the east



(Gong et al., 2014; Cohen et al., 2017). In Asia, the cyclones and fronts are most dominant synoptic weather to loft air pollutants from the PBL to MLT (Liu et al., 2003; Cooper et al., 2004; Ding et al., 2009; Ding et al., 2017), and this kind of synoptic weather are particularly frequent in spring (Chen et al., 1991; Jacob et al., 2003). A number of efforts have been conducted to observe the concentration and composition of mixed air pollutants in the MLT in Asia and to simulate the pathway of continental air masses (Jacob et al., 2003; Huebert et al., 2003; Liu et al., 2003; Hsu et al., 2003; Lee et al., 2014).

However, most of the existing studies focused on the Northwest Pacific region, and less works has been conducted in coastal region of eastern China, a transition area between the region with intense regional pollution and the downwind ocean area. As one of the largest city clusters in the world, the Yangtze River Delta (YRD) region, locates in the most south tip of the polluted northern and eastern plains in coastal eastern China. In spring, the YRD region is generally downwind of the plains area in the north, but also has intense anthropogenic emission because of huge amount of fossil fuel combustion (Ding et al., 2013a, b). The unique geographic location makes this region an ideal place to study the transport, mixing and feedback of dust and anthropogenic pollutants before them were transported over the Pacific Ocean. In this study, we integrated field measurements and numerical simulations for a unique case in March 2015 to investigate mixing of Asian dust, BB and anthropogenic aerosol and its meteorological feedback in middle-low troposphere in eastern China. We describe the data and methods in section 2, present the results of the observations and simulations of the dynamic structure and transport mechanisms for this case, and discuss the meteorological feedback and the environmental impacts of this case in section 3. Finally, summaries are given in section 4.

## 2. Data and method

### 2.1 Data

To investigate the pollution characteristics and validate model's performance, several observational data were utilized in this study. The mass concentrations of $PM_{2.5}$, $PM_{10}$ and $PM_{2.5-10}$ (with aerodynamic diameter less than 2.5, 10 microns, and ranging from 2.5 and 10 microns, respectively) and hourly mass concentrations of inorganic ions ($SO_4^{2-}$, $NO_3^-$, $Cl^-$, $NH_4^+$, $Na^+$, $K^+$, $Ca^{2+}$, $Mg^+$) in $PM_{2.5}$ were measured at a supersite in downtown Nanjing by the Jiangsu Key Laboratory of Environmental Engineering. Descriptions on the site and instrumentation were given in details in Zhou et al.(2016). Meanwhile, aerosol optical depth (AOD) satellite retrievals by moderate resolution Imaging Spectrometer (MODIS, MYD08_D3) were employed to illustrate the spatial patterns of aerosol. Moreover, hourly air quality index (AQI) was acquired through the online access to ambient air monitoring data publicly released by the Ministry of Environmental Protection of the PRC (http://www.zhb.gov.cn/), and were used to analyze the regional $PM_{10}$ and $PM_{2.5}$ characteristics and validate the corresponding simulations. In addition, the CALIPSO level 1 aerosol profiles and vertical feature of aerosol subtype were provided to investigate vertical distribution and transport of mixed aerosols in MLT (Winker et al., 2009). Radiosonde observations from the Integrated Global Radiosonde Archive (IGRA), which are performed at 08:00 and 20:00 Local Time (LT, note that throughout this paper the time refers to LT, unless UTC is specially mentioned), are collected and compared with NCEP



(National Centers for Environmental Prediction) global final analysis (FNL) data to investigate meteorological responses to aerosol pollutions.

## 2.2 Numerical simulations

To investigate the vertical distribution and meteorology feedback of mixed air pollutants in the MLT, we conducted numerical simulations using the Weather Research and Forecasting model coupled with Chemistry (WRF-Chem), which is a three-dimensional Eulerian chemical transport model considers the feedback between meteorology and chemical processes (Grell et al., 2005). In this work, the WRF-Chem version 3.6 was run in a domain with 186×162 grids and a horizontal resolution of 20 km. The model has 30 vertical layers extended from the ground surface to 50 hPa pressure level, with a much higher density in the lower atmosphere. The initial and boundary conditions of meteorological fields were provided by the 6-h NCEP FNL data on 1°×1°grids. The simulations were run from 11 to 26 March 2015, with the first 8-day simulation as spin-up. We performed two groups of simulations. The first group aims to discuss the distribution and sources of the three layers mixed pollutants in the MLT over the eastern China, based on four parallel experiments: 1) with all emission (EXP1), 2) no anthropogenic CO emission from eastern China (EXP2), 3) no dust emission (EXP3), and 4) no BB emission from Indochina (EXP4), to disentangle the individual contribution of transport and mixing processes of different sources. The other group contains two parallel simulations: with and without aerosol radiative feedback (EXP_WF and EXP_WoF), to understand the impact of air pollution on meteorology. BB emission was obtained from Global Fire Emissions Database Version 4.1 (GFED4s) and MIX database (http://www.meicmodel.org/dataset-mix) was used as anthropogenic emission for the present WRF-Chem modelling. Similar model configurations applied in previous works proved a good performance in reproducing the pollution variation in East Asia (Ding et al., 2016; Liu et al., 2016).

Lagrangian particle dispersion modelling was conducted to research the transport mechanisms of air pollutants for the case using the FLEXPART model (Stohl et al., 2005). This model has been widely used in many studies (e.g. Stohl et al., 2002; Ding et al., 2009). The model was also driven by the FNL data. For each target air mass, 4000 particles were released and backwardly run for 7 days. We calculated the averaged residence time of particles of a layer of 100 m above ground surface as the retroplume to investigate the possible impact of surface anthropogenic emission on the target air mass (Ding et al., 2009; Ding et al., 2013c). We also calculated the averaged vertical cross-section of the residence time along the main transport pathways to understand the 3-dimensional structure of air pollution transport and dispersion (Ding et al., 2015). We also used the Hybrid Single-Particle Lagrangian Integrated Trajectory (HYSPLIT) model (Stein et al., 2015) to calculated single-particle trajectories based on Global Data Assimilation System data.





## 3 Results and discussions

### 3.1 Surface observations on anthropogenic aerosols and dust storm

In late March of 2015, Nanjing was influenced by Asian dust, biomass burning and anthropogenic aerosol concurrently. The temporal variation of PM and its chemical compositions measured in the downtown Nanjing demonstrated a multi-day episode

of particle pollution, with maximum of $PM_{2.5}$ and $PM_{10}$ occurred in 19 March and 23 March, respectively (Figure 2). In the first few days of this period, the $PM_{2.5}/PM_{10}$ ratio was generally higher than 0.7, and the secondary water soluble inorganic compositions like $NO_3^-$, $SO_4^{2-}$ and $NH_4^+$ were the main contributors to $PM_{2.5}$ mass concentrations. Apparently, the haze episodes before 23 March were attributed to secondary aerosol formation mainly from anthropogenic pollution. However, the $PM_{2.5}/PM_{10}$ ratio started to drop as low as about 0.2 in the afternoon of 23 March when the $PM_{10}$ reached up to the maximum

of 261 μg m$^{-3}$ but $PM_{2.5}$ remained less than 50 μg m$^{-3}$, indicating possible influence from wind-blown dust. In the meantime, the mass concentration of $Ca^{2+}$, a tracer of soil-derived dust, showed a peak value over 6 μg m$^{-3}$, while secondary inorganic compositions $NO_3^-$, $SO_4^{2-}$ and $NH_4^+$ did not show an obvious synchronous change. Such variation patterns of different aerosol composition further confirm strong impacts from dust in Nanjing after 23 March. Interestingly, a synchronous small peak of $SO_4^{2-}$ (around 12 μg m$^{-3}$) could be observed as the dust plume approached. During existing field campaigns, sulfate-coated

dust particles were often observed during the long-range transport of dust storm, which has been proven to be caused by heterogeneous uptake on mineral dust (Levin et al., 1996; Song et al., 2005). In Nanjing, we observed enhanced precursor oxidation during dust storm in the late spring of 2012 (Xie et al., 2015). The result in this work further demonstrated that secondary sulfate formation was promoted when dust storm mixed with anthropogenic pollution. It highlights that in YRD, which is an FF emission intensive area and the downwind area of Asian dust, the mixing of Asian dust and anthropogenic $SO_2$

may lead to complex chemical transformations and then increasingly polluted dust particles coated by anthropogenic sulfate, posing a significant impact on the regional-scale atmospheric composition and oceanic biogeochemical cycle.

The in-situ measurement gave us a clue that air pollution in Nanjing transferred from anthropogenic FF dominant to dust dominant around 23 March, accompanied with distinct changes in physicochemical properties of particle. To clearly identify the source region and the air mass transport pattern before, during and after the dust event, we calculated 3-day backward

trajectories, starting at the altitudes of 10 m and 1000 m over Nanjing at 08:00LT of 22-24 March respectively using the HYSPLIT model. As indicated by the weather charts together with the backward trajectories for these three days in Fig. 3, before the dust event (i.e. 22 March), a surface high pressure dominated the Mongolia Plateau with its tongue extended to eastern China. The air mass in Nanjing transported very slowly from the plain area with high anthropogenic emission in the northeast (see the trajectory in red). On 23 March, when the high pressure moved southeastward and dominated the continental

area, air masses at the ground surface in Nanjing showed a transport pathway from the YRD city clusters. However at the altitude of 1000 m air masses were originated from the Gobi Desert and swept the North China Plain before recirculating from the ocean, as demonstrated in Fig. 3e. Although dust particle is usually long-range transported in the upper air, it could stretch down to ground and mix with near-surface anthropogenic pollution through daytime convection. This transport pathway and





mixing mechanism was confirmed by the fact that coarse particle concentration at the ground surface rapidly increased in Nanjing at noontime on 23 March, similar as the observations during dust storms reported by Xie et al. (2015) and Nie et al. (2014). On 24 March, when the continental high pressure moved eastwardly with separated centers located in coastal eastern China and Northeast China, the backward trajectories at 10 m and 1000 m altitudes showed a much faster transport from the

Northeast China through the ocean.

Spatiotemporal variations of observed hourly concentrations of $PM_{2.5}$ and $PM_{2.5-10}$ by the monitoring network of Ministry of Environmental Protection in YRD were adopted to examine the spatial evolution of this multi-day episode in the YRD region. As shown in Fig. 4, the YRD region was influenced by a regional haze events on 21 March, with $PM_{2.5}$ over 120 μg m$^{-3}$ in coastal cities in eastern YRD (Fig. 4a), however, $PM_{2.5-10}$, i.e. the coarse particle shows a different distribution pattern,

with only moderated concentration in the northern part of the domain (Fig. 4d). On 22 March, the concentration of coarse particle in the northern part of the domain enhanced substantially (Fig. 4e) but the $PM_{2.5}$ plumes further moved southerly, showing a contrast spatial distribution in the south and north. The spatial pattern was separated by a cold front extend from the northwest to southeast YRD (See also Fig. 3b). Behind the cold front, i.e. in the north part, high-concentration dust aerosols were carried by the cold front from north, however, the strong wind speed weaken the surface $PM_{2.5}$ pollution by transporting

them southwardly or by lifting them to high altitude. In the front of the cold front, a calm condition favours the accumulation and formation of secondary PM from anthropogenic sources. Here the contrast distribution of $PM_{2.5}$ and dust on both sides of the cold front implies that at ground surface mixing of dust of aged secondary PM in the surface is limited because of their different locations to the cold front. On 23 March, when the cold front moved further southerly, $PM_{2.5}$ in the entire YRD region further decrease but the coarse particle enhanced in the middle YRD, especially an axis from Nanjing to Shanghai (see big

black dots in Fig. 4f), corresponding to the surface $PM_{10}$ maximum on that days (Fig. 2).

To clearly investigate the transport of the dust storms, we collected the ground-based measurements of $PM_{2.5}$ and $PM_{10}$ at several air-quality monitoring stations from north to south along the mainly transport pathways, including Shijiazhuang (SJZ), Jinan (JN), Nanjing (NJ), Shanghai (SH) and Hangzhou (HZ) (the geographic locations are shown in Fig. 5a). The time series of $PM_{2.5-10}$ in Fig. 5b indicated that all the stations were influenced by the dust storm during 22-23 March with a rapid increase

in coarse particle concentration. Temporally, dust covered northern stations like SJZ and JN on 22 March and arrived at the YRD on the following day. Interestingly, not all the dust occurrence at these sites showed a time lag from northwest to southeast. For example, on 22 March, the occurrence of dust peak at JN was even earlier than that at SJZ. On 23 March, the occurrence of dust peak at NJ, the most northwest city in the YRD region, was the latest among the three YRD cities. These spatiotemporal variations were caused by different transport characteristics between these sites. Forward trajectory for the peak hours at these

cities shows that the dust plumes from the northern cites less directly transported to YRD but to the ocean area from north of Jiangsu province. The non-lagged transport pathway indicates that the transport of dust to lower latitude were not by horizontal advection following the ground surface, but most probably transport in high altitude and then influence the ground surface by other processes, such as the vertical mixing due to the development of PBL height in the daytime. This phenomenon has been



observed in Xie et al. (2015), in which the surface measurements did not show any obvious signal of dust until dust in the upper PBL was mixed downwards to the surface.

## 3.2 Vertical source attribution of mixed pollutions

As aforementioned, due to substantial influence from synoptic processes, this aerosol pollution episode featured mixing of multiple emission sources and significant vertical heterogeneity, which played an important role in regional-scale air quality in YRD. Synoptic fronts usually extend from the surface up to the middle troposphere. Air pollutants can rise along the circulations ahead of cold fronts to the MLT, where pollutants can be transported further and have longer atmospheric residence time, leading to multiple pollution layers in vertical. Based on the analysis of the surface measurement above, there may exist several aerosol layers in the YRD region during this case. A vertical cross-section of aerosol subtype along the CALIPSO satellite track by observation at 02:04 LT on 23 March 2015 (Fig. 6a) showed us a direct vertical distribution picture of the mixed aerosols above YRD. In the ground level the regional polluted continental aerosols mainly accumulated by the local anthropogenic emissions mixed with polluted dust. The dust aerosols took control above the PBL to the altitude about 5 km. The CALIPSO satellite track depicted the transport pathway of mineral dust from higher altitude in North China to downwind areas. It noteworthy that there was a smoke plume at the top of the dust layer in the south of Shanghai, formed a huge pollution belt at the altitude of 5 km and extended about 10 latitudes above the South China Sea. To discuss the transport pathway and source contribution of mixed pollutants in different layer, a matrix of 7-day backward trajectories started at 10 m, 2 km and 5 km above ground level were applied along the CALIPSO satellite track, as illustrated in Fig.7. According to the trajectories started at the ground level, we found that air masses near southeast coast of China were mainly influenced by FF emission-intensive regions in eastern China, including both YRD and the North China Plain. Comparatively, the backwards trajectories at the altitude of 2 km indicated the air mass in southeast coast of China mainly originated from the Gobi and Taklimakan deserts through long-distance transport. The clean marine aerosol located at about 20ºN was from the South China Sea, then circulated through the southeast urban area and mixed with anthropogenic pollutants and dust. The air mass near the altitude of 5 km was mainly came from Indochina with intensive BB emissions in March (van der Werf et al., 2006). It is clear that during this case the potential source regions of different vertical layers varied significantly.

In order to clarify the potential source of MLT aerosol over the YRD region which was identified as polluted dust according to CALIPSO retrievals, a matrix of 3-day backward trajectories was applied from YRD region started at 5 km above ground level. The air masses were originated from the ground surface of inland area in the southwest, with trajectory recirculated in a small area between Hunan and Hubei Province more than one day, then travel eastward a distance at hundreds of kilometres and reached YRD at a high level of 5 km. Pollutants in the MLT have a longer lifetime for there is no deposition and colder temperatures (Zhou et al., 2013). The FLEXPART model was used to further illustrate the different modes of transport. The particles were released starting at 00:00LT 23 March. The particles are released at the height of about 5 km over YRD region (red square in fig. 8b). The air mass was derived from the boundary layer to the middle troposphere for about 12h.



When it reached the middle troposphere the air mass turned more easterly. The 3-day backward surface distribution (<100m) of particles released in YRD was mainly concentrated on the locations of the southwest inland region. According to this large number of backward particles, we isolated those representing the Warm Conveyor Belts (WCBs) by applying criteria from Eckhardt et al. (2004), which requires 2-day air masses to travel eastward a distance at least 10 longitudes, northward exceeding 5 latitudes, and vertically at least 50 % of the average tropopause height at the ending position. The tropopause height in YRD region was about 11-12 km, yielding a 2-day ascent criterion of more than 5 km. Applying these criteria isolated most of YRD region that were influenced by the WCBs. Ding et al. (2009) also reported the WCBs could transport the megacities plumes in the North China Plain to mid-troposphere over the Northeast Asia.

To understand the evolution of the cold front and its impact on vertical structure of atmospheric aerosol, we conducted simulations using regional chemical transport model WRF-Chem. By comparing with available observations, the model is proven capable of reproducing the variations of meteorological fields and $PM_{10}$ distributions in YRD region (Table 1). Fig. 9 demonstrated the vertical cross-section of simulated $PM_{2.5-10}$, anthropogenic CO and BC, wind field and perturbation potential temperature along coastal East China at 08:00 LT on 23 March. The position of frontal surface, which was identified by the mutation of the perturbation potential temperature and wind field, was marked in red line in Fig. 9c. The dust and CO dominated behind the frontal surface and could transport equatorward and eastward with the evolution of synoptic systems. The high concentrations of CO above the frontal surface was transported from inland ground and lifted by the frontal system, as discussed above. Dust can rise along circulations ahead of cold fronts. Behind cold fronts, pollution tends to be low near the ground, mineral dust can be lofted in mesoscale wind systems (Huang et al., 2010). The dust mixed with anthropogenic pollution in middle troposphere (6-10 km) behind the frontal surface. BC showed a high concentration ahead of the frontal system, which was mainly due to the tremendous emission from BB in Indochina and subsequent transport along with the westerlies in the low troposphere. Thus, the BB pollutants could mix with anthropogenic and dust pollutants at the junction of the front. These vertical structure characteristics caused by frontal system were clearly described by the model results and satellite observations.

To attribute and analyse source of the three layers mixed pollutants in the MLT over eastern China, we performed four parallel simulations, experiments with all emission (EXP1), no anthropogenic CO emission from eastern China (EXP2), no dust emission (EXP3), and no BB emission from Indochina (EXP4). Fig.10 shows the mixed pollutants from dust ($PM_{2.5-10}$), anthropogenic emission (CO) and BB (BC) at 5 km altitude and vertical cross-sections of mixed pollutants averaged from the black box area at 08:00 LT 19, 21 and 23 March. Eastern China is one of the world's most FF emission-intensive area with fast industrialization and urbanization, and CO is a good tracer tracking transport of FF pollution in the troposphere with a life time of weeks to months. During the early stage of event on 19 March, the greatest level of surface CO (>400 ppbv) was mostly concentrated in Eastern China. With the development of frontal system, CO was gradually lofted from the ground surface to the altitude of 5-6 km on 23 March. Spatial pattern of CO at the altitude of 5 km in Fig. 10(a-c) give a clear picture of how CO was elevated by cold frontal system. As the CO-concentrated air masses reached the middle troposphere, it began to transport



eastward, evidently demonstrated by an area with CO concentration exceeding 300 ppbv in the south YRD in the early morning on 23 March. At the same time, the dust storm which was accompanied by cold fronts, swept over eastern China from 21 to 25 March. This dust event mainly originated from Taklimakan Desert due to extremely high wind shear, and was transported with westerly wind in the free troposphere and then approached YRD on 21 March. On that day the dust layer suspended at

5 the altitude of 4-6 km. Increasingly stronger vertical mixing and deposition process brought the dust down to the surface (Fig. 10f), whereby substantial dust signals like extremely $PM_{2.5}/PM_{10}$ ration and $Ca^{2+}$ concentration peak were detected at the ground station in Nanjing (Fig. 2). Furthermore, March features most intensive forest fires in Indochina (Jacob et al., 2003; Zhou et al., 2013; Lin et al., 2014; Dong et al., 2015). Both CALIPSO and WRF-Chem simulation indicated that during this case BB was pumped along the mountain area and transport to the south of YRD region and even the East China Sea by the

10 westerly. BB pollution influenced YRD on 23 March and mainly concentrated around 4-6 km (Figs. 10c and 10f). Obviously, the FF pollutants and dust mixed well in large area of eastern China from ground level to low troposphere, and staggered with BB aerosol belt in coastal regions of southeast China in the free troposphere.

Asian dust and inland anthropogenic emissions can be transported to the eastern China though both slow transport near surface and relatively fast transport in the MLT, thereby influencing regional air quality. Eastern China is under influence of

15 the typical Asian Monsoon, and the long-range passage of cold front associated with Asian Winter Monsoon, could cause lots accumulated air pollutants and dust plumes are easily to be transported to downwind coastal city clusters of east and south China (Mori et al., 2003; Huang et al., 2010; Liu et al., 2011; Xie et al., 2015; Zhang et al., 2016; Ding et al., 2017), and further mixed with BB pollutants transported from Indochina. The degradation of air quality could pose an adverse effect on human health in eastern China. Under control of East Asian Monsoon, the climatological occurrence of fronts has suggested a high

frequency of the pollution lifting events in East China, and in YRD regions in winter and early spring.

### 3.3 Meteorological Feedback of mixed pollutants

As discussed above, the frontal system lifted multiple-source pollutants up and resulted in multiple aerosol layers in the MLT. Meanwhile it is noteworthy that both dust and BB aerosols feature high light-absorbing efficiency, which certainly exerts substantial impacts on radiation transfer and regional climate. To shed more light on aerosol-MLT meteorological feedback,

we conduct two parallel simulations with (EXP_WF) and without (EXP_WoF) aerosol radiative effects. Spatial distributions of particle and temperature response to their radiative effect at different altitudes on 23 March 2015 are shown in Fig. 11. At the ground surface, light scattering and also absorption due to multi-layer aerosol resulted in substantial dimming effect. The spatial pattern of surface dimming was generally consistent with those of pollution. On the other hand, upper-air heating was found at both altitudes of 4 km and 8 km (Figs. 11b-c). The 4-km warming mainly stretched along the coastal southeast China,

correspondent with BC from BB in Indochina transported to the coastal southeast China and staggered with FF and dust aerosol belt in north of coastal regions. Meanwhile, after long-range transport, windblown dust reaching the downwind YRD suspended aloft and the influence of dust on temperature would be stronger with several-day heating accumulation (Liu et al.,



2016), which efficiently heated the surrounding air near 8 km. Previous modelling and observation studies has emphasized the importance of upper-air aerosols in changing temperature stratification because of more incident solar radiation and less efficient vertical heat exchange (Samset et al., 2014; Ding et al., 2016; Wang et al., 2018). In this case, BB and dust aloft caused by frontal system were expected to exert important role in changing the air temperature in MLT. The reduced ground
surface temperature and heating trend in the upper air could jointly favor the accumulation of pollutants in the boundary layer (Ding et al., 2016; Petäjä et al., 2016) and loft of dust in the atmosphere (Huang et al., 2015; Liu et al., 2016).

To further understand the role of dust and polluted aerosols in the aerosol-MLT meteorological feedback, we compared air temperature between FNL reanalysis data and radiosonde observations in Anqing and Shantou (locations are marked in red circle in Fig. 11f) during this case in Fig. 12. Without considering the aerosols' feedback, the NCEP FNL Operational Global
Analysis data minus the observation data could be utilized to evaluate the strength of the meteorological feedback caused by aerosol. Overall, the FNL temperature profiles were basically consistent with the sounding observations. However, at the crucial layers with significant dust or BB pollution, FNL temperature showed a negative bias. Specifically, an obvious underestimation (2.3 ℃) was apparent at 500 hPa compared with sounding data in Anqing. As mentioned above Anqing was under control of mixed pollutants uplifted by the frontal system and transport by the westerly streamline, suggesting an
important role of polluted aerosols in changing the air temperature in the middle troposphere. While from 900 hPa to 700 hPa a substantial underestimation (1.2-2.9 ℃) was found in Shantou, indicating the temperature increase influenced by the vertical distribution of the BB aerosols in the low troposphere transported from South Asia as shown in Fig. 12b.

Cross-section of averaged pollutants from FF, BB and dust and corresponding temperature perturbations along 115 °E and 120 °E from 22 to 24 March were shown in Fig.13. As shown the FF and dust pollutants dominated in the whole PBL, and
lifted by the cold front to the free troposphere in the mid-latitude area. While the BC calculated from BB emission in South Asia lifted by the terrain followed by eastward transport in the free troposphere, mainly distributed between 15-30 °N. Since the lifted pollutants especially the BC and dust aerosol could efficiently warm the free troposphere, the MLT temperature change diagnosed by the EXP_WF and EXP_WoF averaged during these three days displays similar pattern with pollution distribution in Fig.13. BC in the MLT and high concentrations of FF and dust pollutants accumulated in the PBL could lead to
stronger warming. The 115 °E cross section mainly passes through eastern China's urban agglomerations, and the FF and dust pollution concentrations are higher than the 120 °E section of the eastern ocean. The warming of the inland section at 15-30 °N and 2-8 km height is even more pronounced, especially at an altitude of 2-4 km, which is related to the higher concentration of BB aerosols over South China. On the eastern surface of the ocean, the temperature increase at 30-45 °N in the ground level of the eastern ocean is stronger, which is related to more apparent warming effect of FF and dust aerosol on the ocean. All
these proved that the mixed pollutants and their perturbation on meteorology could transport eastward to the Pacific.

Meteorological parameters are very important factors influencing routine air quality. The transport and dispersion of air pollutants was strongly affected by the synoptic weather and boundary layer dynamics. Meanwhile high concentration of certain aerosols could change meteorological parameters, synoptic dynamics and even climate change through their strong



radiative feedback (Hansen et al., 1997; Li et al., 2011; IPCC, 2013; Huang et al., 2015).Our previous works found how the air pollution-boundary layer feedback changed the weather and in turn how the modified meteorological conditions enhanced the accumulation of air pollution (Ding et al., 2013a; Ding et al., 2016; Liu et al., 2016; Ding et al., 2017; Wang et al., 2018). This two-way interaction between air pollution and meteorological conditions was widely discussed in the PBL (Fan et al.,

2015; Gao et al., 2015; Wang et al., 2015; Wilcox et al., 2016). In this case, we proved this interaction could happen in the MLT due to the special vertical distribution and transport of the mixed pollutants.

## 4. Summaries and implications

Ground-based measurements, satellite observations and numerical simulations with Lagrangian dispersion model and chemical transport model are combined to investigate vertical structure, transport characteristics, source contribution, and

meteorological feedback for a unique multi-day pollution episode in YRD in March 2015, which was characterized by concurrent influence from multi-sources like fossil fuel combustion, biomass burning, and dust emission. In situ measurements in the YRD cities showed that this episode started with high concentration of $PM_{2.5}$, especially secondary inorganic compositions $NO_3^-$, $SO_4^{2-}$, and $NH_4^+$, and then changed into dust pollution dramatically. The secondary fine particulate pollution near surface at the beginning was mainly influenced by the local anthropogenic emission from fossil fuel combustion

sources, and the dust plumes were mainly originated from the Taklimakan and Gobi Deserts in Northwest China and transported by cold fronts. The high concentrations of pollutants above the frontal system was transported from inland ground and lifted by the front and the dust mixed with anthropogenic pollution in middle troposphere above and also behind the frontal surface. Biomass burning plumes in Indochina uplifted through the terrain, transported by westerlies and staggered with anthropogenic and dust aerosols in coastal regions of southeast China in the free troposphere. The three pollutant sources could

all be transported to eastern China, especially the YRD region around the latitude of 30°N, caused a structure of multi-layer pollutants and well mixed pollutants there. These solar absorption aerosols, like black carbon and dust, could cause a net heating effect in the MLT and cooling effect in PBL in different quantity from south to north along the coastal China, which further caused the changes in the dynamics in the troposphere, especially in the PBL. All these processes could be summarized as a schematic figure in Fig. 14.

Spring has been considered as a season with strong continental outflow in Asia. The "polluted" outflow will have great impact on the atmospheric environment and regional even global climate. What is emphasized in this study is that large-scale synoptic weather like cold fronts in spring played an important role in uplifting not only anthropogenic fossil fuel combustion sources but also in transporting naturally emitted dust and biomass burning aerosols. East Asia has experienced fast rapid industrialization and urbanization in the last decades, leading to intensive emission of fossil fuel combustion pollutants.

Simultaneously, this region is also under the influence of Asian dust storm and biomass burning in Southeast Asia in spring. Therefore, multiple-source and multilayer air pollution with distinguished emission sources can be therefore frequently





observed in spring with periodic cold front activities. The mixed pollutants could experience complex chemical and physical processes in the atmosphere and subsequently modify the structure of atmospheric components and also meteorological parameters over the downwind areas, such as the Pacific Ocean. Unfortunately, existing field measurements in this region are predominately surface measurements. More emphasis ought to be paid on vertical characterization of pollution so as to get a

5 better understanding of the vertical structure, transport and feedback mechanism of mixed pollutants pollution and their associated climate and environmental effects in the Asian region with intense human activities and strong monsoon.

**Acknowledgements**

The study was supported by National Key Research & Development Program of China (2016YFC0200500), National Science

10 Foundation of China (D0512/91544231, D0512/41422504), National Science and Technology Support Program (2014BAC22B02) and (2014BAC22B05), Air Pollution Control Technology Support Program of Jiangsu Province (BM2012063), Science and Technology Support Program of Jiangsu Province (SBE2014070928).




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



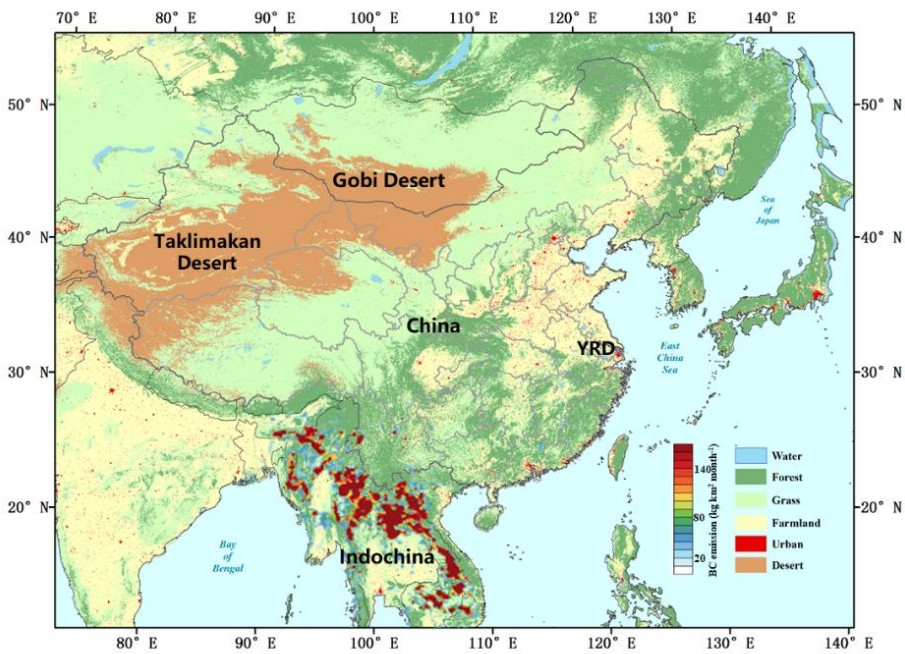

**Figure 1.** Map showing the land cover and averaged carbon emission from biomass burning in March in East Asia. Note: the Land cover data were the 2012 MODIS Land Cover Type product and biomass burning emission data were from GFED4 emission inventory.

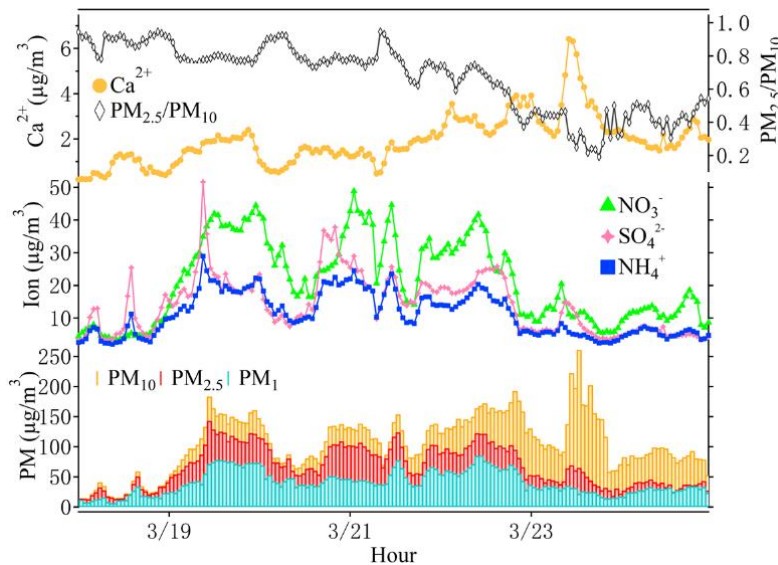

**Figure 2.** Time series of concentrations of $PM_{10}$, $PM_{2.5}$, $PM_1$ and main water soluble ions of $PM_{2.5}$ ($NO_3^-$, $SO_4^{2-}$, $NH_4^+$, $Ca^{2+}$) measured in Nanjing during 18-23 March 2015.



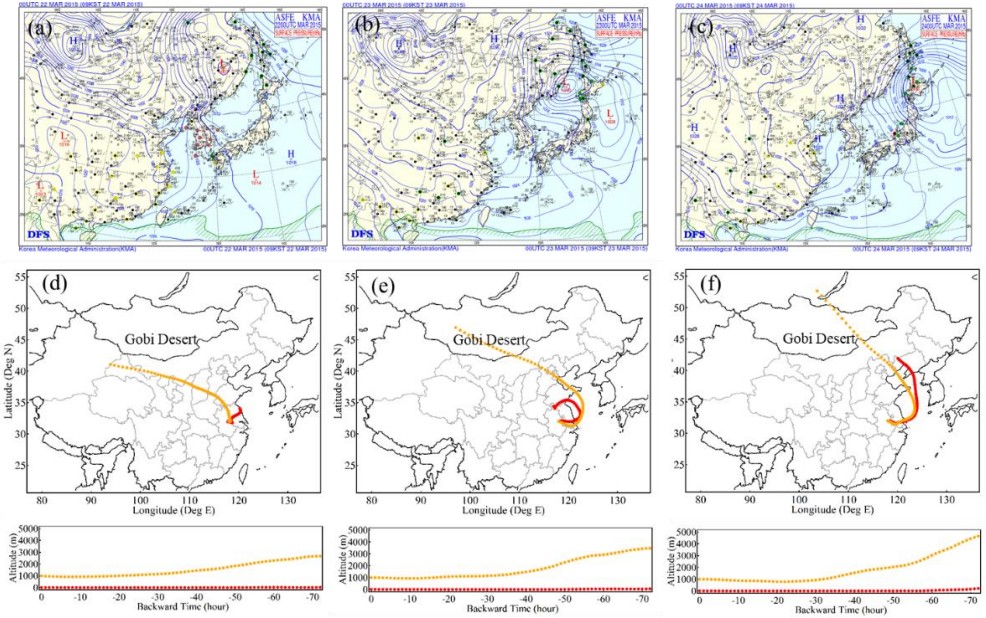

**Figure 3.** Weather charts and 3-day backward trajectories starting at altitudes of 10 m and 1000m over Nanjing at 08:00LT on **(a) (d)** 22, **(b)(e)**23, and **(c)(f)** 24 March 2015, respectively.

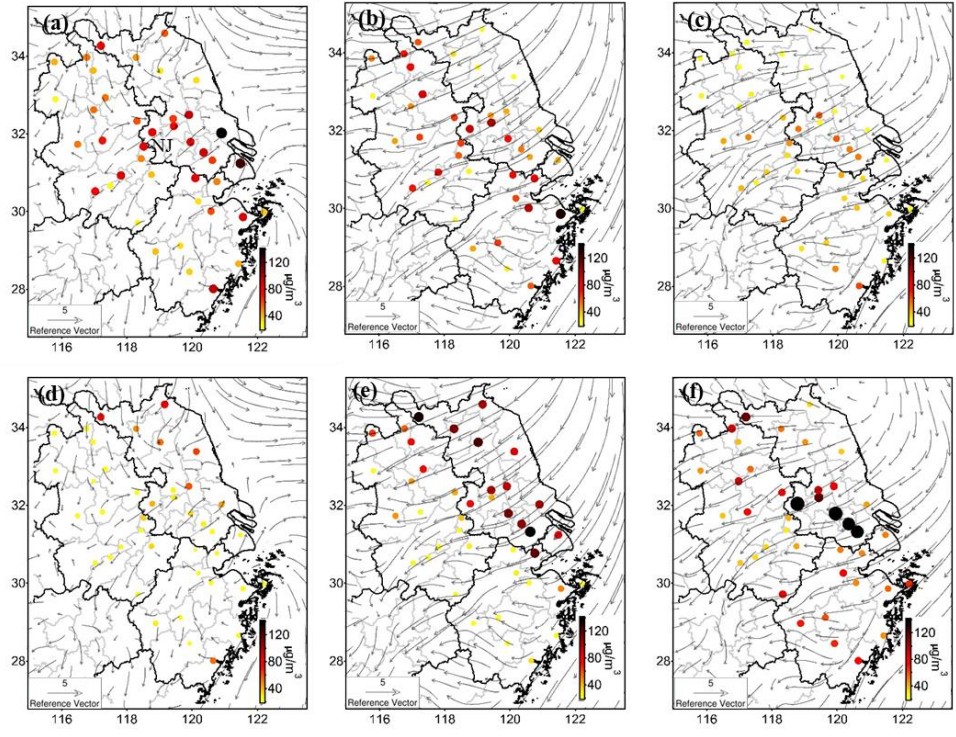

**Figure 4.** Spatial distributions of wind and observed hourly concentrations of $PM_{2.5}$ **(a-c)** and $PM_{2.5-10}$ **(d-f)** over the Yangtze River Delta at 12:00LT on 21, 22 and 23 March 2015, respectively.





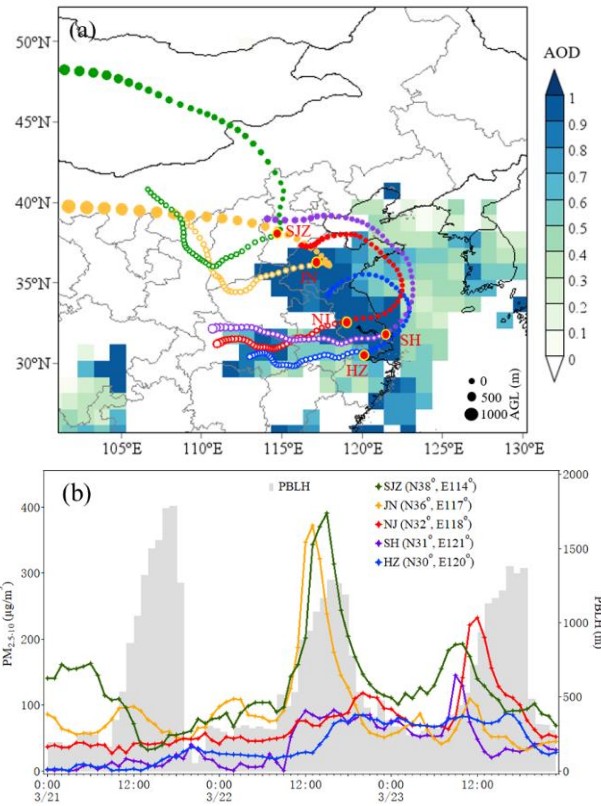

**Figure 5. (a)** AOD distribution on March 22 and 72-hr backward (solid circle) and forward (hollow circle) trajectories at 08:00 23 March,
2015 and **(b)** time series of PM$_{2.5\text{-}10}$ and averaged PBL height at 5 cities in the eastern China during 21-23 March 2015.

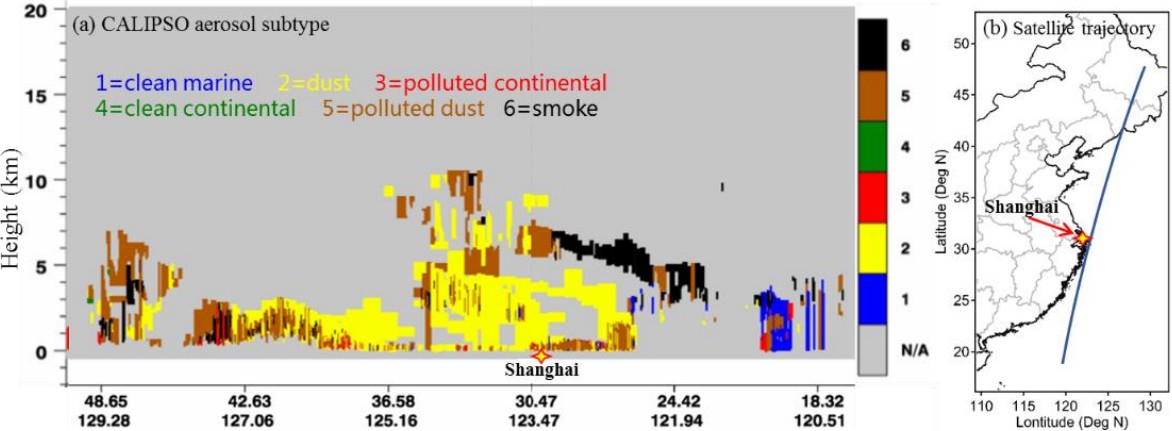

**Figure 6. (a)** Vertical cross-section of aerosol subtype along the **(b)** CALIPSO satellite track at 02:04 LT 23 March 2015.



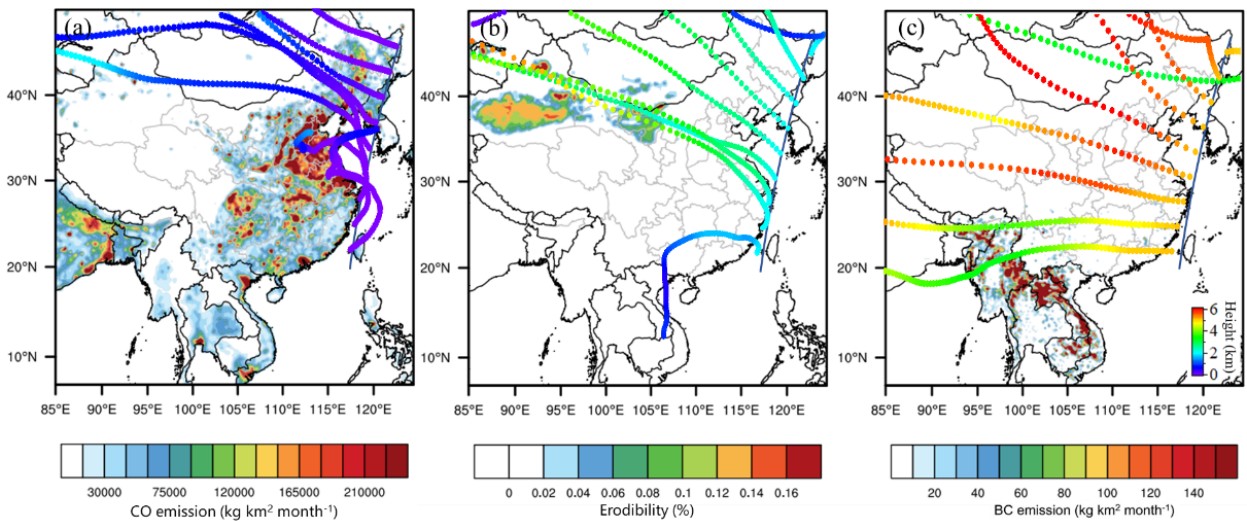

**Figure 7.** 7-day backward trajectory at different altitudes **(a)** 10 m, **(b)** 2000 m, and **(c)** 5000 m above ground level along the satellite orbit. Note: Anthropogenic carbon monoxide emission, soil erodibility and BC emission from biomass burning were shown respectively on these maps.

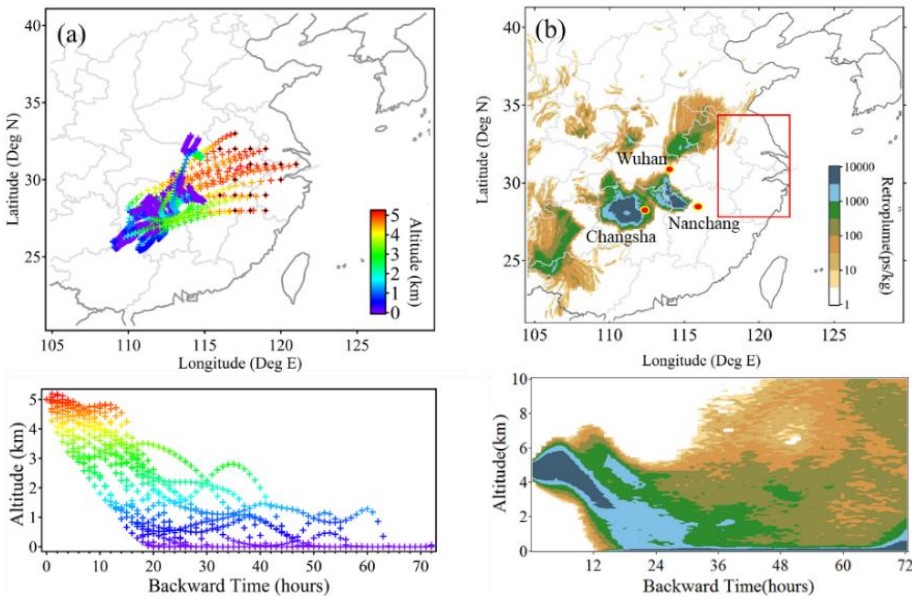

**Figure 8**. **(a)** 3-day backward trajectories and **(b)** averaged surface retrolume of particles released in the red square at 00:00 LT 23 March 2015. The lower panels give the vertical distribution of trajectories or particles.




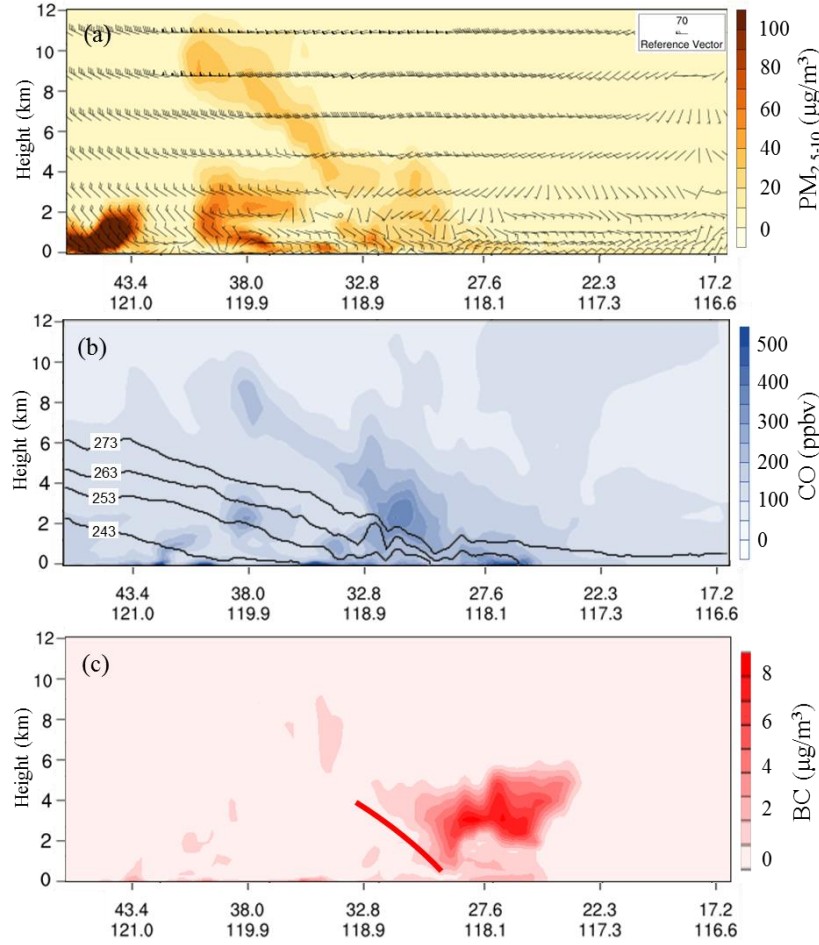

**Figure 9.** Vertical cross-section of **(a)** PM$_{2.5-10}$ and uv wind field (m/s) **(b)** CO and potential temperature (contour lines, Unit: K) and **(c)** BC concentration obtained from WRF-Chem simulation along the coastal eastern China on 08:00 LT 23 March 2015.





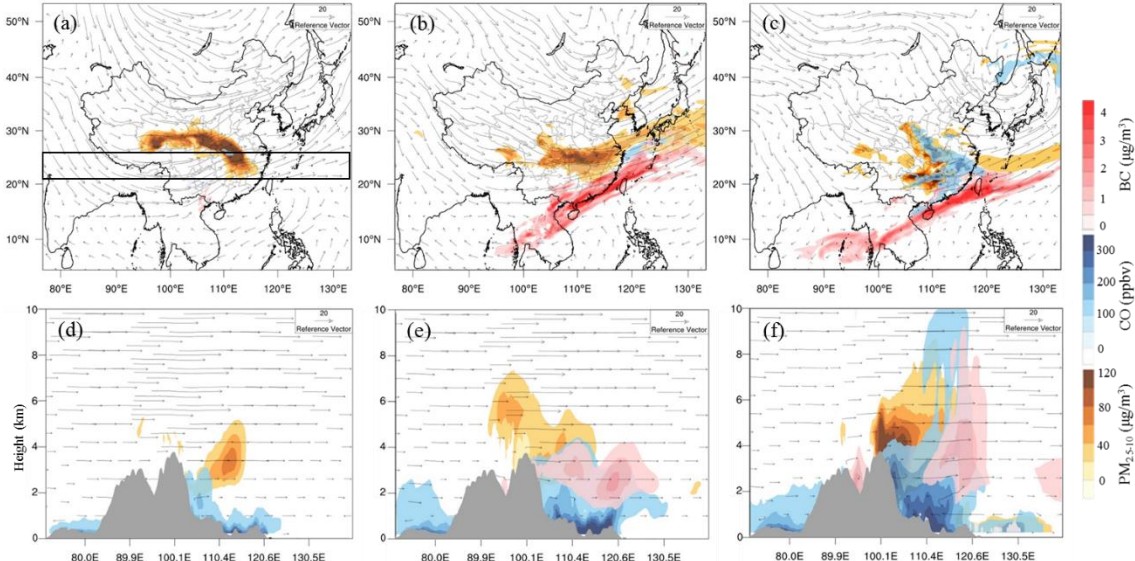

**Figure 10.** WRF-Chem simulated mixed pollutants from dust (PM$_{2.5-10}$), anthropogenic emission (CO) and BB (BC) at 5 km altitude and vertical cross sections of mixed pollutants averaged from the black box area in Fig. 10a at 08:00 LT on **(a)(d)** 19 March **(b)(e)** 21 March, and **(c)(f)** 23 March.

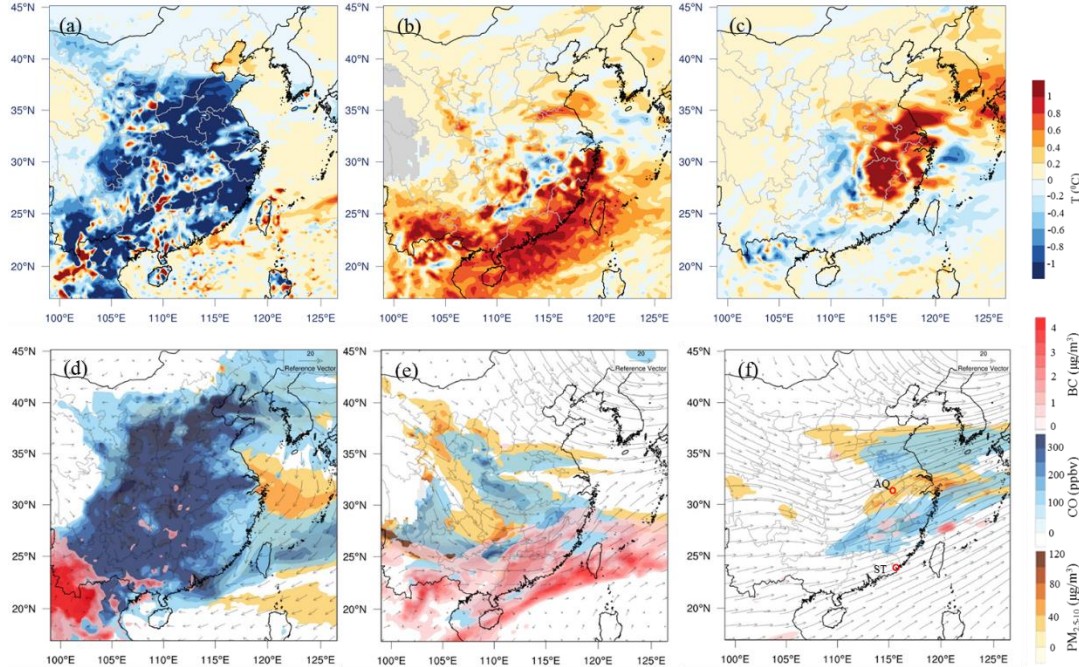

**Figure 11.** Spatial distributions of modelled air temperature change due to the meteorological feedback of mixed pollutants and pollutants distribution at **(a)(e)** ground surface **(b)(e)** 4 km altitude, and **(c)(f)** 8 km altitude at 12:00 LT 23 March 2015.



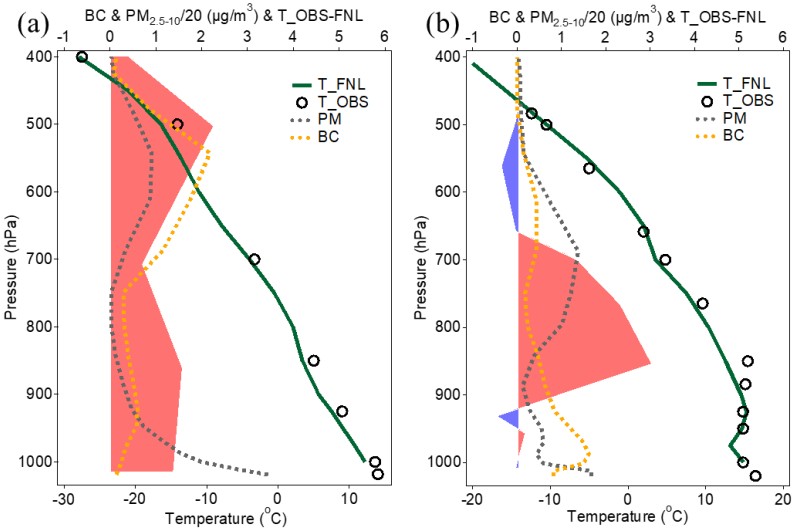

**Figure 12.** Vertical profiles of air temperature from NCEP FNL Operational Global Analysis data, the corresponding radiosonde observations (black circles), temperature difference between observation and the FNL data (filled red and blue colour to zero), calculated BC and PM$_{2.5-10}$ for (a) Anqing (30.5 N, 117.1 E) and (b) Shantou (23.3 N, 116.7 E) at 20:00 LT 23 March 2015.

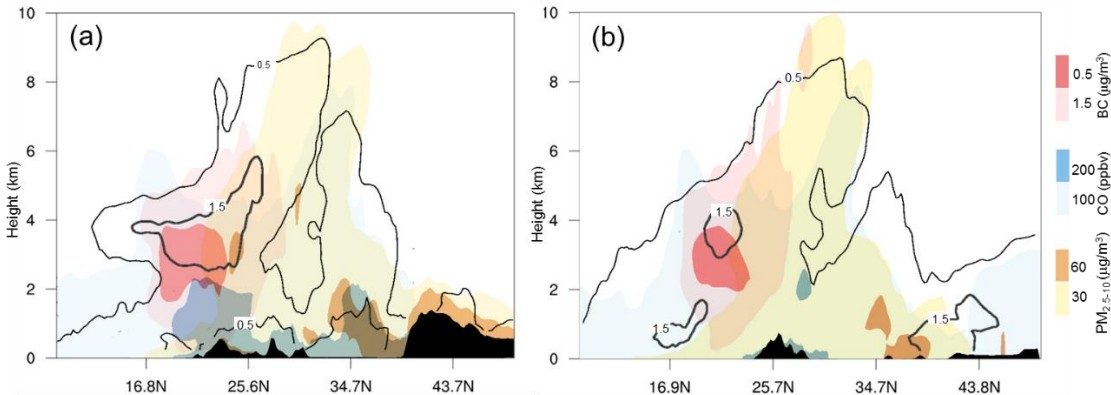

**Figure 13.** Cross-section of averaged pollutants from FF (CO), BB (BC) and dust (PM$_{2.5-10}$), and air temperature change diagnosed by EXP_WF and EXP_WoF ( C) on the longitude of **(a)** 115 E **(b)** 120 E during 22-24 March 2015.



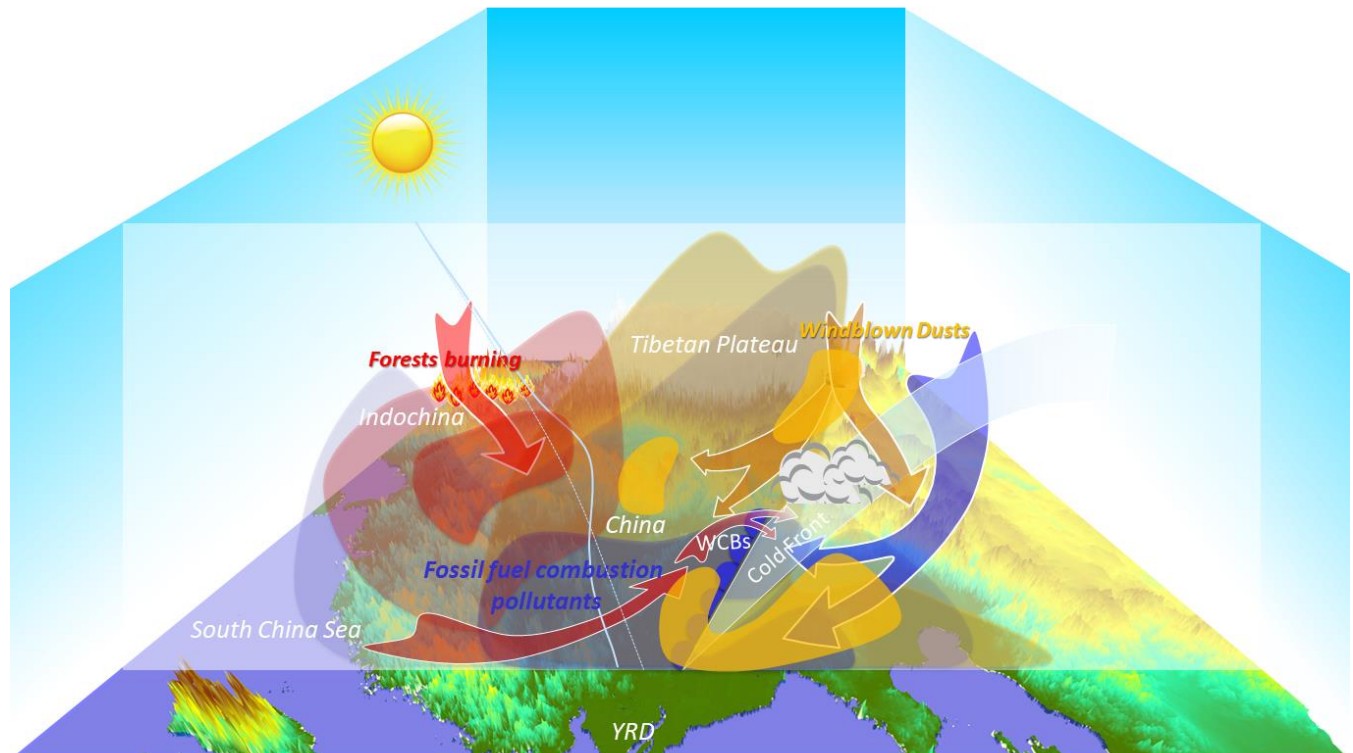

**Figure 14.** A schematic figure for the transport, mixing and feedback of dust, biomass burning and fossil fuel combustion pollutants in eastern Asia.

5    **Table 1.** Statistical analyses of the simulated meteorological variables and PM$_{10}$ versus the ground observations. MB and RMSE refer to mean bias and root-mean-square error respectively.

| Stations | Index | 2-m temperature (°C) | Wind speed (m s$^{-1}$) | PM$_{10}$ (μg m$^{-3}$) |
|----------|-------|----------------------|--------------------------|---------------------------|
| NJ | MB | -0.35 | 0.01 | -2.30 |
|    | RMSE | 1.41 | 0.70 | 13.26 |
| SH | MB | -0.28 | -0.04 | 1.80 |
|    | RMSE | 1.93 | 0.92 | 15.16 |