# Peer review of "Transport, mixing, and feedback of dust, biomass burning and anthropogenic pollutants in eastern Asia: A case study"

_Atmospheric Chemistry and Physics, 2018_

## Referee Comment (RC1) · Anonymous Referee #2 · 19 Jul 2018

Based on observations and model simulations, the authors examined transport, mixing, and feedback of dust, biomass burning and anthropogenic pollutants in late March 2015 over the Yangtze River Delta (YRD) region. They found that fossil fuel aerosols (FF) mainly accumulated near surface mixed with dust and biomass burning aerosol (BB) from the Southeast Asia were transported by westerlies around the altitude of 3 km. They also found solar absorption aerosols from FF, BB and dust could cause significant feedback with MLT meteorology. The topic is interesting and could contribute to the current knowledge of aerosol pollution in eastern China. However, there are still some issues need to be addressed before it can be considered for publication.

My major concern is the inconsistency of proxies for FF, BB and dust between observation and model. For the observational data, the authors used concentrations of $PM_{2.5}$ and $PM_{2.5-10}$ as proxies as FF and dust in Figures 2 and 4, which is fair. But in the following part, the authors introduced BB, which is also in $PM_{2.5}$ and $PM_{2.5-10}$. In addition, the author only used data for SO4, NO3, NH4, Ca2+ without BC and OC. For the model simulations, the authors did simulations with all emission, no anthropogenic CO emission from eastern China, no dust emission, and no BB emission from Indochina. They used CO as proxy of FF and only turned it off in eastern China. First, CO is not a good proxy for FF/$PM_{2.5}$. Second, turn FF off in eastern China includes information of both source sector and source region, which should be analyzed separately. I suggest conducting simulations as with all emission, no anthropogenic emission (all FF aerosol and precursor, e.g., $SO_2$, $NH_3$, NOx, BC, OC), no dust emission, and no BB emission. This will give a clearer result and then FF $PM_{2.5}$ (sum of $SO_4$, $NH_4$, $NO_3$, BC, OC) and BB $PM_{2.5}$/BC can be used as proxies for model.

Minor comments:

Method mentioned AQI but not used.

Page 8 Line 11: 'In the ground level the regional polluted continental aerosols mainly accumulated by the local anthropogenic emissions mixed with polluted dust.' What does this mean? Aerosols are mainly contributed by local anthropogenic FF emissions and dust?

Page 10 Line 1: 'CO concentration exceeding 300 ppbv in the south YRD in the early morning on 23 March'. Why CO showed high value here (proxy of FF/PM2.5) and PM2.5 showed low value in Figure 4c

Page 11 Line 20: 'BC calculated from BB emission in South Asia'. How did the authors calculate this value? Or BB emission in Southeast Asia/Indochina?

Both surface measurement, satellite data and model simulation have uncertainties. The authors should discuss these uncertainties and the potential influences to the results in this study.

Some other studies also examined sources and transport of anthropogenic aerosols in China (e.g., Yang et al., 2017, 2018) and feedbacks of dust on monsoon meteorology (e.g., Lou et al., 2017). The authors may give credit to these studies.

Reference:

Yang, Y., H. Wang, S. J. Smith, P.-L. Ma, and P. J. Rasch, Source attribution of black carbon and its direct radiative forcing in China, Atmos. Chem. Phys., 17, 4319–4336, doi:10.5194/acp-17-4319-2017, 2017.

Yang, Y., H. Wang, S. J. Smith, R. Zhang, S. Lou, Y. Qian, P.-L. Ma, and P. J. Rasch, Recent intensification of winter haze in China linked to foreign emissions and meteorology, Sci. Rep., 8, 2107, doi:10.1038/s41598-018-20437-7, 2018.

Lou, S., L. M. Russell, Y. Yang, Y. Liu, B. Singh, and S. J. Ghan (2017), Impacts of interactive dust and its direct radiative forcing on interannual variations of temperature and precipitation in winter over East Asia, J. Geophys. Res. Atmos., 122, 8761–8780, doi:10.1002/ 2017JD027267.

---

## Referee Comment (RC2) · Anonymous Referee #3 · 22 Jul 2018

The study by Zhou et al. reported very interesting events with a vertical mixture of dust, biomass burning and anthropogenic pollutants in eastern Asia (i.e., Nanjing-Shanghai, Yantze-River-Delta region in China). The transport, mixing and feedback to the regional meteorological conditions have been comprehensively discussed with the support of different modelling tools and observational data. This study contributes to the current understanding of the air pollution formation in YRD and highlights the need of comprehensive vertical observations in the polluted city clusters in east China. Since previous studies in this region were mostly based on ground-based measurements, the inclusion of vertical structure analysis provided further insight of the distinct pollution regimes. Thus, I recommend publication of this study after the following issues have

been addressed.

P5 L10: How about the chemical initial and boundary conditions? Please specify the configurations.

P5 L10-15: Which dust scheme has been used here? How was the performance compared to observations? I suggest to split the statistics in Table 1 into anthropogenic dominated period and dust dominated period, and a time series of model vs observation would be helpful.

P6 L10-15: Please clarify during the dust event if the 'secondary inorganic compositions NO3-, SO42- and NH4+ did not show an obvious synchronous change (with the increasing concentration of Ca2+)' or 'a synchronous small peak of SO42- . . . could be observed as the dust plume approached'. These two statements sound contradictory. Also, I would suggest the author to mark the 'synchronous' peak of SO42- and Ca2+ in Fig. 2.

P6 L15-20: How was the relative humidity during the studied period? If it is wet chemistry, was it similar 'foggy' conditions like in Xie, Ding et al. (2015) or an aerosol water/haze mediated chemistry as in Cheng, Zheng et al. (2016)? Otherwise, was it more of a heterogenous uptake and oxidation on the dry particles or a new particle formation enhanced by the dust events, e.g., Nie, Ding et al. (2014), which may not be coated-sulfate on dust particles then?

P10 L25: What is the refractive index of dust treated in WRF-Chem simulation?

P11 L5-P12 L5: How frequent does such kind of vertical mixture of dust, anthropogenic pollution and biomass burning occur in YRD? What is the effect of vertical mixed structure? Here the two examples, one demonstrates the effect of elevated polluted aerosols (Anqing) and the other one is for the effect of biomass burning aerosols (Shantou). Will the meteorological feedback effect (per unit of mass) increase when all three pollutants mixed together? Here the authors show that the warming and dimming effects

can change the vertical temperature profile at Anqing and Shantou. I am wondering after including the direct radiative forcing of aerosols in the WRF-Chem simulation (EXP_WF) whether the model simulated pollutant fields (e.g., PM2.5, PM2.5-10, inorganic ions, CO, or organic etc.) agreed better with observations than the EXP_WoF case?

P11 L5: Here the authors mainly referred to the previous studies about the effect of the reduced ground surface temperature and heating in the upper air. I would suggest to have more evidences and discussion here or later (P11 L20-30) with the difference of cross-section of averaged pollutant from FF, BB and dust with or without aerosol direct radiative effect as the air temperature change diagnosed by EXP_WF and EXP_WoF in Fig. 13.

P11 L15: It is interesting that the warming peak (red shaded between 800-900 hPa) at Shantou does not co-located with the peaks of PM2.5-10 and BC at around 700 hPa (Fig. 12b). It would be great if the authors could further comment on it.

P11 L20: I agree with the other referee that the author should demonstrate here if it is appropriate to use CO, BC and PM2.5-10 as surrogates of FF, BB and dust, respectively. I would suggest the authors to analyze the difference between the base case EXP1 and the scenario simulations EXP2 (no anthropogenic CO emission in eastern China), EXP3 (no dust emission) and EXP4 (no BB emission from Indochina) and show the contributions to CO, BC and PM2.5-10 in these two cross-sections (in percentage) from the three types of emissions (i.e., anthropogenic CO emission in eastern China, dust emission, BB emission from Indochina).

P19 Fig. 2: The label of x-axis should be 'Date' (maybe indicate that 'hourly' data are showing here in the figure caption). I suggest to tick the full date range from 3/18 to 3/25 on the x-axis.

P24 Fig. 11: It should be '(a)(d)' in figure caption.

**References**

Cheng, Y., et al. (2016). "Reactive nitrogen chemistry in aerosol water as a source of sulfate during haze events in China." Science Advances 2(12).

Nie, W., et al. (2014). "Polluted dust promotes new particle formation and growth." Scientific Reports 4: 6634.

Xie, Y., et al. (2015). "Enhanced sulfate formation by nitrogen dioxide: Implications from in situ observations at the SORPES station." Journal of Geophysical Research: Atmospheres 120(24): 12679-12694.
* * *

---

## Author Comment (AC1) · 3 Oct 2018

**Response to Referee #2**

*Based on observations and model simulations, the authors examined transport, mixing, and feedback of dust, biomass burning and anthropogenic pollutants in late March 2015 over the Yangtze River Delta (YRD) region. They found that fossil fuel aerosols (FF) mainly accumulated near surface mixed with dust and biomass burning aerosol (BB) from the Southeast Asia were transported by westerlies around the altitude of 3 km. They also found solar absorption aerosols from FF, BB and dust could cause significant feedback with MLT meteorology. The topic is interesting and could contribute to the current knowledge of aerosol pollution in eastern China. However, there are still some issues need to be addressed before it can be considered for publication.*

**Response:** We would like to thank the referee for the overall encourage comments and providing the insightful suggestions, which indeed help us further improve the manuscript.

*My major concern is the inconsistency of proxies for FF, BB and dust between observation and model. For the observational data, the authors used concentrations of $PM_{2.5}$ and $PM_{2.5-10}$ as proxies as FF and dust in Figures 2 and 4, which is fair. But in the following part, the authors introduced BB, which is also in $PM_{2.5}$ and $PM_{2.5-10}$. In addition, the author only used data for SO4, NO3, NH4, Ca2+ without BC and OC. For the model simulations, the authors did simulations with all emission, no anthropogenic CO emission from eastern China, no dust emission, and no BB emission from Indochina. They used CO as proxy of FF and only turned it off in eastern China. First, CO is not a good proxy for FF/$PM_{2.5}$. Second, turn FF off in eastern China includes information of both source sector and source region, which should be analyzed separately. I suggest conducting simulations as with all emission, no anthropogenic emission (all FF aerosol and precursor, e.g., SO2, NH3, NOx, BC, OC), no dust emission, and no BB emission. This will give a clearer result and then FF $PM_{2.5}$ (sum of $SO_4$, $NH_4$, $NO_3$, BC, OC) and BB $PM_{2.5}$/BC can be used as proxies for model.*

**Response:** We used different proxies for FF, BB and dust in this work. The $PM_{2.5}$ and $PM_{10}$ concentration measured at air quality monitoring stations were used to explain surface observations on regional transport of anthropogenic aerosols and dust storm.

Actually, there is no signal on BB based on direct ground-based measurements since BB plume was above 3 km over the coastal area. Thus in Figure 2, we did not include BC and OC concentrations. However, according to the observation of CALIPSO vertical source attribution of mixed pollutions, we found the layer of BB from Indochina in the MLT. To better understand the vertical structure of atmospheric aerosol during this period, we conducted simulations using regional chemical transport model WRF-Chem. We agree the simulation design as suggested (with all emission, no anthropogenic emission, no dust emission, and no BB emission), which is exactly what we have done in the current work. In the simulation without anthropogenic emission, we did turn off all FF aerosol and precursor rather than just CO. We just used CO to identify the spatial patterns of anthropogenic pollutions. For clarity, we added table 2 to illustrate the simulation design of this work and rephrased the corresponding descriptions. Please refer to Page 5 Line 15, Page 9 Line 25 and Page 26 Line 5 in the revision.

*Minor comments:*

*1) Method mentioned AQI but not used.*

**Response:** The $PM_{2.5}$ and $PM_{10}$ concentration data were used, and the relevant description has been modified in the revised manuscript.

*2) Page 8 Line 11: 'In the ground level the regional polluted continental aerosols mainly accumulated by the local anthropogenic emissions mixed with polluted dust.' What does this mean? Aerosols are mainly contributed by local anthropogenic FF emissions and dust?*

**Response:** We have rephrased it to 'In the ground level the polluted dust aerosols mainly accumulated by the local anthropogenic emissions mixed with dust.'

*3) Page 10 Line 1: 'CO concentration exceeding 300 ppbv in the south YRD in the early morning on 23 March'. Why CO showed high value here (proxy of FF/PM2.5) and PM2.5 showed low value in Figure 4c*

**Response:** As demonstrated in Figure 10f,11d, the high value of CO in surface layer (greater than 300 ppb, 300 ppb is not high in spring time relatively) was in the west outside the YRD region and the value of CO. However in Figure 11d showed relatively

high value of CO in the city cluster from Suzhou to Nanjing, and in Figure 4c also showed relatively high value of $PM_{2.5}$ concentration in the same region. The model results and observations are consistent in the distribution pattern of the surface FF pollution. While in Figure 10c the CO concentration only exceeding 100ppbv and at the altitude of 5km, and also the YRD region is clean. We have revised the misleading description and please refer to page 10 line 1.

4) *Page 11 Line 20: 'BC calculated from BB emission in South Asia'. How did the authors calculate this value? Or BB emission in Southeast Asia/Indochina?*

**Response:** Accepted. In the revised manuscript, we changed to 'BC calculated from BB emission in Indochina'. Please see Page 11 Line 20-21

5) *Both surface measurement, satellite data and model simulation have uncertainties. The authors should discuss these uncertainties and the potential influences to the results in this study.*

**Response:** In this case, we proved the special vertical distribution and transport of the mixed air pollutants and dust could happen in the MLT. The characteristics of mixed pollution and source contribution in three layers of MLT over eastern China were generally reproduced by our model simulations, which is consistent with the ground observation and satellite data. However, existing field measurements in this region are predominately in-situ surface measurements. The model results in the troposphere still lack sufficient quantitative analysis. More emphasis ought to be paid on vertical characterization of pollution so as to get a better understanding of the vertical structure, transport mechanism, and more specific research on the thermodynamic and dynamic energy process and impact area of this phenomenon was needed in the future study.

**Response to Referee #3**

*The study by Zhou et al. reported very interesting events with a vertical mixture of dust, biomass burning and anthropogenic pollutants in eastern Asia (i.e., Nanjing-Shanghai, Yantze-River-Delta region in China). The transport, mixing and feedback to the regional meteorological conditions have been comprehensively discussed with the support of different modelling tools and observational data. This study contributes to the current understanding of the air pollution formation in YRD and highlights the need of comprehensive vertical observations in the polluted city clusters in east China. Since previous studies in this region were mostly based on ground-based measurements, the inclusion of vertical structure analysis provided further insight of the distinct pollution regimes. Thus, I recommend publication of this study after the following issues have been addressed.*

**Response:** We would like to thank the referee for providing the insightful suggestions, which indeed help us further improve this work.

1) *P5 L10: How about the chemical initial and boundary conditions? Please specify the configurations.*

**Response:** The chemical initial and boundary conditions were MOZART-4 results acquired from National center for Atmospheric research (NCAR). This information has been added in the Section 2.2 in the revised manuscript.

2) *P5 L10-15: Which dust scheme has been used here? How was the performance compared to observations? I suggest to split the statistics in Table 1 into anthropogenic dominated period and dust dominated period, and a time series of model vs observation would be helpful.*

**Response:** We adopted the Noah land-surface scheme and the Monin-Obukhov surface layer scheme (Alizadeh Choobari et al., 2012) and GOCART dust emission module (Ginoux et al., 2001) in this study. The source function based on soil erodibility by wind was shown in Fig. 7b. Similar model configurations applied in previous works proved a good performance in reproducing the dust pollution variation in East Asia (Liu et al., 2016). Also, the model results agreed well with satellite retrievals on vertical distribution of various aerosol types in Fig. 6, which verify the model's capacity on reproducing the regional dust transport in the whole troposphere. To further confirm the

model performance on dust concentration reproduction, we added comparison between simulated meteorological and $PM_{10}$ observations in Table 2.

3) *P6 L10-15: Please clarify during the dust event if the 'secondary inorganic compositions $NO_3^-$, $SO_4^{2-}$ and $NH_4^+$ did not show an obvious synchronous change (with the increasing concentration of $Ca^{2+}$)' or 'a synchronous small peak of $SO_4^{2-}$ ... could be observed as the dust plume approached'. These two statements sound contradictory. Also, I would suggest the author to mark the 'synchronous' peak of $SO_4^{2-}$ and $Ca^{2+}$ in Fig. 2.*

**Response:** Accepted. The description 'secondary inorganic compositions $NO_3^-$, $SO_4^{2-}$ and $NH_4^+$ did not show an obvious synchronous change (with the increasing concentration of $Ca^{2+}$)' was deleted. In addition, 'synchronous' peak of $SO_4^{2-}$ and $Ca^{2+}$ are marked in grey area in the revised Figure 2.

4) *P6 L15-20: How was the relative humidity during the studied period? If it is wet chemistry, was it similar 'foggy' conditions like in Xie, Ding et al. (2015) or an aerosol water/haze mediated chemistry as in Cheng, Zheng et al. (2016)? Otherwise, was it more of a heterogenous uptake and oxidation on the dry particles or a new particle formation enhanced by the dust events, e.g., Nie, Ding et al. (2014), which may not be coated-sulfate on dust particles then?*

**Response:** Cross session of relative humidity and $SO_2$ concentration along the backward trajectory on 23 March in Figure R1 showed relatively dry conditions (RH<30%) in the nearing 10 hours with high value of $SO_2$. We have revised the discussions in Page 6 Line 15-20.

[Figure]

Figure R1 Cross session of relative humidity and SO$_2$ concentration along the backward trajectory on 23 March

5)  *P10 L25: What is the refractive index of dust treated in WRF-Chem simulation?*

**Response:** The refractive index of dust has always been a controversial issue, which depends on the composition and wavelengths. A typical value has been assumed as 1.53 + 0.006i based on several models (Dey, 2004). During the dust events in Beijing from 2004 to 2006, real and imaginary parts of the dust refractive index at 440-1020 nm ranged from 1.52 to 1.56, 0.007 to 0.010, respectively (Wu et al., 2009). Nakajima et al. (1989) suggested the imaginary part of Asian dust to be 0.005 and 0.006, respectively. An averaged imaginary part of 0.007 for solar spectrum was recommended for Gobi desert dust (Stone et al., 2007). According to these recommended value and field observations, we assumed that the short-wave refractive index of 1.55 + 0.006i was acceptable as the refractive index of Asian dust for this case study. The long-wave refractive index of dust was strongly wavelength-dependent, and we adopt the recommended values from the Optical Properties of Aerosols and Clouds dataset (Hess et al., 1998).

6)  *P11L5-P12L5: How frequent does such kind of vertical mixture of dust, anthropogenic pollution and biomass burning occur in YRD? What is the effect of vertical mixed structure? Here the two examples, one demonstrates the effect of elevated polluted aerosols (Anqing) and the other one is for the effect of biomass burning aerosols (Shantou). Will the meteorological feedback effect (per unit of*

*mass) increase when all three pollutants mixed together? Here the authors show that the warming and dimming effects can change the vertical temperature profile at Anqing and Shantou. I am wondering after including the direct radiative forcing of aerosols in the WRF-Chem simulation (EXP_WF) whether the model simulated pollutant fields (e.g., PM$_{2.5}$, PM$_{2.5-10}$, inorganic ions, CO, or organic etc.) agreed better with observations than the EXP_WoF case?*

**Response:** The polluted dust associated with cold fronts could extend to the troposphere and transported from inner Asian continent to south coastal region in spring. While the BB from Indochina dominated in the whole middle-low troposphere over South China coastal region. So we believe this kind of vertical mixture could occur in YRD. Dong et al. (2018) also reveal the co-existence condition that dust from the Taklimakan and Gobi Desert and biomass burning from Peninsular Southeast Asia can reach to the west Pacific region simultaneously in boreal spring by alpine observation.

The effect of such vertical mixed structure is relatively difficult to define, especially under the influence of the cold front large-scale system. When the solar radiation is strong during the day, ground dimming over eastern China and upper-air heating was found at both altitudes of 4 km over Coastal South China and 8 km over YRD region (Figs. 11a-b-c), which corresponding to mixed pollutants of different layers.

The model simulated pollutants fields with meteorological feedback in PBL were usually agreed better with observations in the ground level during heavy pollution period on previous studies. In this study we extend the feedback throughout the troposphere. The model simulated pollutant fields difference needs more contamination enhancement observation of vertical troposphere and further study.

7) *P11 L5: Here the authors mainly referred to the previous studies about the effect of the reduced ground surface temperature and heating in the upper air. I would suggest to have more evidences and discussion here or later (P11 L20-30) with the difference of cross-section of averaged pollutant from FF, BB and dust with or without aerosol direct radiative effect as the air temperature change diagnosed by EXP_WF and EXP_WoF in Fig. 13.*

**Response:** In Fig. 13 we calculated the averaged pollutants and air temperature change diagnosed by EXP_WF and EXP_WoF over land and ocean. In the daytime the ground surface temperature could be reduced, while in the night the polluted aerosols could absorb the energy reflected from the ground and leading to warm effect. As a result the

averaged warm effects extend from ground to middle troposphere. We also calculated the daily change of air temperature and PBL change diagnosed by EXP_WF and EXP_WoF (°C) over YRD region as shown in Figure R2. In the daytime, it showed the same change characteristics and could suppress the development of the PBL. Figure 11 and 12 showed the effects from regional to single site in different layer of MLT, and Figure 13 consider the integrated warming effect of land and sea over the entire pollution period.

[Figure]

Figure R2 Daily change of air temperature and PBL change diagnosed by EXP_WF and EXP_WoF (°C) over YRD region on 23 March.

8) *P11 L15: It is interesting that the warming peak (red shaded between 800-900 hPa) at Shantou does not co-located with the peaks of PM$_{2.5-10}$ and BC at around 700 hPa (Fig. 12b). It would be great if the authors could further comment on it.*

**Response:** The warming in low troposphere over Coastal South China was caused by the BB as shown in Fig. 11b, e, 12b, 13ab. The mountainous terrain in Shantou is complicated, and FNL temperature profiles needs to be interpolated to compare with observation results. The resolution of the model grid points could also cause the error.

9) *P11 L20: I agree with the other referee that the author should demonstrate here if it is appropriate to use CO, BC and PM$_{2.5-10}$ as surrogates of FF, BB and dust, respectively. I would suggest the authors to analyze the difference between the base case EXP1 and the scenario simulations EXP2 (no anthropogenic CO emission in eastern China), EXP3 (no dust emission) and EXP4 (no BB emission from Indochina) and show the contributions to CO, BC and PM$_{2.5-10}$ in these two cross-*

*sections (in percentage) from the three types of emissions (i.e., anthropogenic CO emission in eastern China, dust emission, BB emission from Indochina).*

**Response:** We agree that conducting simulations as with no anthropogenic emission (all FF aerosol and precursor not just CO) and we simulated without any anthropogenic emission in this study. We analyze the difference between the base case EXP1 and the scenario simulations. The misleading descriptions have also been corrected.

10) *P19 Fig. 2: The label of x-axis should be 'Date' (maybe indicate that 'hourly' data are showing here in the figure caption). I suggest to tick the full date range from 3/18 to 3/25 on the x-axis.*

**Response:** Accepted. We have modified it as required.

11) *P24 Fig. 11: It should be '(a)(d)' in figure caption.*

**Response:** Accepted. We have modified it as required.

**Reference**

Dey, S., Tripathi, S. N., Singh, R. P., & Holben, B. N. (2004). Influence of dust storms on the aerosol optical properties over the indo-gangetic basin. Journal of Geophysical Research Atmospheres, 109(D20), -.

Dong, X., Fu, J. S., Huang, K., Lin, N. H., Wang, S. H., & Yang, C. E. (2018). Analysis of the co-existence of long-range transport biomass burning and dust in the subtropical west pacific region. Scientific Reports, 8(1), 8962.

Hess, M. (1998). Optical properties of aerosols and clouds : the software package opac. Bull.am.meteor.soc, 79(5), 831-844.

Nakajima, T. (1989). Aerosol optical characteristics in the yellow sand events observed in may, 1982 at nagasaki-part ii models. Journal of the Meteorological Society of Japan.ser.ii, 67(2), p267-278.

Stone, R. S., Anderson, G. P., Andrews, E., Dutton, E. G., Shettle, E. P., & Berk, A. (2007). Incursions and radiative impact of asian dust in northern alaska. Geophysical Research Letters, 34(34), 623-642.

Wu, Z. J., Cheng, Y. F., Hu, M., & Wehner, B. (2009). Dust events in beijing, china

(2004–2006): comparison of ground-based measurements with columnar integrated observations. Atmospheric Chemistry & Physics Discussions, 9(3), 6915-6932.

**Response to Referee # 1**

The Referee #1 raised some comments during the initial submission stage but didn't submit report during the discussion stage. According to the handling editor's suggestion, here we give a response to his/her comments as bellow.

*The authors have used surface and remotely sensed measurements, and a combination of analytics and some forward models, to try to understand the underlying causes behind a set of highly polluted conditions in 2015. I really like the overall topic studied and the conclusions found. In particular, these are highly supportive of my own independent findings, using different approaches, measurements, and models. In particular, I am impressed with the surface and local measurements provided in Jiangsu Province, and believe that they add a significant value. They also are the strongest part of the paper. I am highly suspicious of the modeling and CALIOP results, and it is for these reasons that I urge the paper to be vastly improved. I hope and believe that after addressing these points, as well as other improvements in the analysis, that this paper will make an excellent contribution to ACP and to the literature.*

**Response:** We would like to thank the comments and suggestions, which help further improve the manuscript.

*First of all, to my knowledge, there is no such biomass burning emissions inventory or urban emissions inventory as they have described with daily resolution over their parts of the world. I have worked hard to create my own, because I could find no other. And the links to data provided in their paper do not provide access to such data.*

**Response:** There has been multiple emission inventories for biomass burning to provide input needed for modeling atmospheric chemistry and air quality in a consistent framework at scales from local to global, some of which do provide daily emission intensity and has been extensively utilized in chemical transport modelling. Daily satellite observations of fire signal are generally to identify the location and time of biomass burning, and thus make it possible to estimate the fire emission with a high spatiotemporal resolution. For instance, Fire INventory from NCAR (FINN) provides daily resolution, global emission estimates from biomass burning at a horizontal resolution of ~1 $km^2$ (Wiedinmyer et al., 2011). Similar daily inventories also include

GFAS and APIFLAME. Some regional/global models also support the online calculation of biomass burning emission based on daily fire satellite retrievals.

*Secondly, the methods used for modeling, by completely turning off some emissions over some region, are not scientifically sound. Although this has been done by others before, there is strong evidence back all the way from Chen and Prinn (2006) and more recently by Cohen and Wang (2014) that this is not mathematically or computationally correct.*

**Response:** Kalman filter method is recommended to estimate contributions from regional sources according to previous work (Chen and Prinn, 2006; Cohen and Wang, 2014). However, it is not suitable for this work. First, Kalman filter is generally applied for long-term data. It should be noted that what we focus here is just a typical pollution case with multiple kinds of emission sources. Additionally, orthogonality of the regions is required for the Kalman Filter equations to be mathematically precise. But in the case, the FF, BB and dust source region are overlapped.

*Thirdly, the model results are highly inconsistent for the BC and CO, against measurements. There is an obvious spatial/temporal mis-match. Furthermore, the concentrations of BC and CO are far too low. Additionally, there is a large inconsistency between BC and PM2.5, even though BC is a component of PM2.5!*

**Response:** More discussions on model evaluation will be added in the revised manuscript. We will conduct comparisons between simulations and corresponding observations for meteorological field and available measurement on particulate matters. We did not compare BC concentration against measurement in this work, which was limited by the availability of the measurements on BC on regional scale.

*Furthermore, the way that CALIOP is used here is in appropriate. The backscatter needs to be considered first and foremost. The errors are already large in this product. The "aerosol type products" from CALIOP should be used with extreme caution. And certainly not a single pass over a single point, especially one that crosses only over the East China Sea, when the ground measurements were found over land more than 100km away.*

**Response:** CALIOP detection used in this work serve as supplementary evidence on

vertically inhomogeneous pollution in this region. It is LPDM model and WRF-Chem simulations that are combined to identified the different emission source. Given the scarcity of vertical observation on atmospheric pollution, we adopted CALIOP retrieval here despite of its uncertainties.

*For these reasons, I can not at the present time believe the modeling results. I am not sure if it is just an emissions error, a resolution error, a physics error in terms of aerosol processing, etc. I also am not able to find confidence in the use of CALIOP, which is unfortunately overly important in their final results. I urge them to work harder and dig more deeply. I look forward to seeing this special episode explored in great detail and making an important contribution to the literature!*

**Response:** We appreciate the referee's comments although we disagree with him/her. However, these comments somehow help us further improve the manuscript.